# A Detailed Comparative Study on Some Physicochemical Properties, Volatile Composition, Fatty Acid, and Mineral Profile of Different Almond (*Prunus dulcis* L.) Varieties

Okan Levent 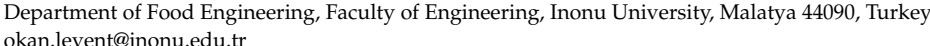

Department of Food Engineering, Faculty of Engineering, Inonu University, Malatya 44090, Turkey;
okan.levent@inonu.edu.tr

**Abstract:** In the present investigation, the main purpose of the research was to reveal the differences among the almond genotypes in terms of their physicochemical properties, volatile composition, fatty acid, and mineral profile. For that reason, ten different almond genotypes originated from different countries were subjected to relevant analysis. The results showed that the total oil, protein, and ash levels of the almond samples ranged between 30.84–41.43%, 17.43–22.72%, and 2.90–3.40%, respectively. Additionally, total phenolic content of the samples was in the range of 38.7–101.03 mg GAE/100 g sample. It was revealed that the almond samples were rich in monounsaturated fatty acids, and oleic acid was the major one with levels of 61.22–77.63%. For all samples, potassium, magnesium, and phosphorus were the major minerals, and the highest concentration was for potassium with levels of 6192.08–11,046.05 mg/kg. Volatile profile analysis showed that the toluene, 4-octanone, pinacol, and 2-methylpentanal were the dominant volatile compounds for all almond genotypes. The results revealed that the different almond varieties showed significant differences depending on the origin.

**Keywords:** almond; aroma compounds; fatty acids; minerals; phenolic

## 1. Introduction

Almond (*Prunus dulcis*) knows as also *Amygdalus communis,* which is the most widely cultivated nut crop of Mediterranean area and shows many benefits for human health, is a tree species that together with peach is included in the subgenus Amygdalus. In the world, almond is one of the most consumed nuts, especially in high-income economies, and it was reported that the total production of almonds in the whole world was more than two million tons. In the production of almonds, USA was the first country with more than one-million-ton production for 2020, according to the records of FAOSTAT. It is one of the most important nut species that has increased production and consumption ratios day by day. Karaat [1] stated that the almond-growing areas started to increase in the last decades because of an increased demand for almond due to its natural and healthy food ingredients. Şimsek [2] informed that the almond is a good material in the diet due to providing micronutrients, macronutrients, and various bioactive constituents. Jenkins et al. [3] also stated that the almond kernel is used to treat the heart and autoimmune system, rheumatoid arthritis, and cancer in recent years. Barreca et al. [4] reported that the nut has many nutritious ingredients such as fatty acids, lipids, amino acids, proteins, carbohydrates, vitamins, and minerals, as well as secondary metabolites. Almonds contain lipids (around 50%), proteins (around 25%), and carbohydrates (around 20%), and have a low moisture content and diverse minor bioactive compounds. The beneficial effects of almond consumption are associated with its composition of macro- and micronutrients [5].

Almond has an acceptable and desired aroma and flavor, and it is accepted as a nutritious and delicious fruit due to its protein, fat, mineral matter, fibre, and vitamin E content [6]. Almond proximate composition is generally influenced by the by ecological

conditions, location, and technical and cultural practices [7,8]. There are different investigations performed for the characterization of the different almond genotypes. Beyhan et al. [6] investigated the fatty acid compositions of some important almond (*Prunus amygdalus* L.) varieties selected from Turkey, and they reported that there was no huge significance among the samples. Kodad et al. [9] studied the variability of oil content and major fatty acid of the different almond genotypes and reported that the differences were significant. Kalita [2018] reviewed the cardiovascular effects of almond, and they reported that the almonds have a functional role for the regulation cardiovascular problems. In addition, Mathpal and Rathore [2021] reported the health benefits of almonds, and they informed that almonds are highly rich in vitamin E, copper manganese, fiber magnesium, phosphorus, monounsaturated fatty acids, and riboflavin protein, among other nutrients.

In the current study, the aim was to characterize the different almond varieties originated from different countries (Ferragnes, Ferraduel, Marta, and Lauranne from France; Texas and Nonpareil from USA; Guara from Spain; Yaltinskii from Russia; and Nurlu and Acıbadem from Turkey) in terms of their main proximate composition, fatty acid and mineral profile, and volatile compound composition. Additionally, a cluster analysis was performed to compare the almond genotypes and to determine the similar varieties by their characterized properties.

## 2. Materials and Methods

### 2.1. Materials

The different almond genotypes were provided from Agricultural Credit Cooperative Adıyaman Kahta Almond-Pistachio Enterprise (harvest year 2021). During the harvest year, the average temperature for the growing area was 29.99 °C, the lowest temperature was in the range of −2.2 and 16 °C in December–January, and the highest temperature ranged between 22–41.7 °C for July and August. In addition to that, monthly total precipitation ratios were 34.25 mm, while the highest precipitation level was 199.8 mm in January and the minimum precipitation was zero in July. Average monthly total sunbathing time was 283 h. The provided almond varieties were Ferragnes (FS), Ferraduel (FL), and Marta (MA) (Spanish); Lauranne (LU) (France); Texas (TX) and Nonpareil (NL) (USA); Guara (GU) (Spain); Yaltinskii (YS) (Russia); and Nurlu (NU) and Acıbadem (BA) (Turkey). The kernel mass values of the almond genotypes were in the range of 1.88–4.89 g. The highest kernel was for the genotypes of Ferraduel, and the average kernel size was 3.02. After harvesting, they were kept at room temperature for about 1 month, and the provided samples were stored at 4 °C until the related analysis.

### 2.2. Methods

2.2.1. Analysis of Proximate Composition

For the proximate composition, total oil, total protein, and total ash contents of the almond samples were determined according to Nizamlıoğlu and Nas [10]. For the determination of total oil content of the samples, the powdered almond samples were exposed to extraction by n-hexane for approximately 6 h using the Soxhlet extraction system. At the end of the duration, the oil content was calculated depending on the mass balance. For the determination of the total protein content of the samples, Kjeldahl protein analysis methodology was followed, and the determined nitrogen content was multiplied by 6.25 to calculate the final total protein content of the samples. Total ash level of the almond samples was determined as following: for this aim, the powdered almond kernels were exposed to carbonization at firstly 200 °C for 24 h and then 6 h at 600 °C. At the end of the carbonization, the ash levels of the samples were calculated according to the mass balance [1]. All measurements were performed as replicates with three repetitions.

2.2.2. Total Phenolic Content Analysis

For the extraction of the almond samples, 0.5 g of powdered almond was mixed with 30 mL of ethanol (96%) and stirred for 30 min using a magnetic stirrer (Clifton, Nickel

Elctro MSH1, Thermo Fisher Scientific, UK). At the end of the mixing, the suspension was filtered using a Whatman no. 1 filter paper, and the filtrate was exposed to concentration until 10 mL left as a volume using rotary evaporator (Bibby RE 100, By Bibby Sterilin Ltd., Stone, UK) at 40 °C. The prepared final solution was used as the almond extract for the total phenolic content analysis. To determine the total phenolic content of the samples, the Folin–Ciocalteu reagent method was followed as suggested by Yıldırım et al. [11] and Durmaz and Alpaslan [12]. For this purpose, 100–500 µL of the samples was diluted to 1 mL using ethanol, and this volume was adjusted to 46 mL with incorporation of distilled water. Then, 1 mL of the Folin–Ciocalteu reagent was added, and this mixture was left to stand for 3 min. After that, 3.0 mL of sodium carbonate (2% $w/v$) was added to the mixture, the mixture was mixed well by vortex and the tubes were incubated at room conditions for 120 min. At the end of the duration, the absorbance of the samples was recorded at 700 nm using a spectrophotometer (UV-1700, Shimadzu, Kyoto, Japan). To calculate the total phenolic content of the samples, a calibration curve was prepared by using gallic acid standard at different concentration, and the results were given as mg gallic acid equivalent (GAE)/100 g sample. All measurements were performed as replicates with three repetitions.

### 2.2.3. Analysis of Mineral Composition of the Samples

Macro and micro element compositions of the almond samples were determined according to the suggested methods by Mertens [13,14]. Totally, 250 mg of powdered almond sample was exposed to digestion process by using 4 mL of HNO3 (65%) in a speed wave microwave digestion equipment (Berghof, MSW-4, Eningen, Germany), and the digested samples were subjected to elemental analyses. Contents of P, Mg, Ca, K, Zn, Mn, Cu, Fe, and Na were determined according to standard stock solution (1000 mg/L) of each element using an inductively coupled plasma mass spectrometry (ICP-MS) (NexION 350X, Perkin Elmer, Waltham, MA, USA). All measurements were performed as replicates with three repetitions.

### 2.2.4. Fatty Acid Profile of Almond Oil Samples

The fatty acid composition of the almond sample oil was determined according to the method described by Karaat [1]. At first, the almond oil samples were subjected to methylation process. For this process, 100 mg of oil sample was weighed and 100 µL of 2 M KOH (prepared with methanol) was incorporated into oil for saponification and mixed well by a vortex for a while. Then, 3 mL of n-hexane was added to the mixture and the mixture was vigorously shaken with a vortex for 1 min, and then the tubes were centrifuged at $2516\times g$ for 5 min at 25 °C. The supernatant was placed into vials and the samples were exposed to GC-FID analysis (Shimadzu, QP2010 ULTRA) equipped with a flame ionization detector and Rtx-5 MS capillary column. The oven temperature of the GC was programmed as follows: 180 °C for 2 min, increased to 200 °C at 2 °C/min, held at 200 °C for a further 10 min, and then increased to 250 °C at 2 °C/min and kept there for 10 min. The injector and detector temperatures were set as 200 and 250 °C, respectively. The carrier gas was helium at a flow rate of 1.5 mL/min. The results were given as percentages of each fatty acid with regard to total oil content. All measurements were performed as replicates with three repetitions.

### 2.2.5. Volatile Profile of Almond Samples

Volatile profile of almond samples was characterized according to the methodology reported by Doguer et al. [15]. For the identification of the volatile compounds of each almond sample, solid phase microextraction (SPME) process was performed by using Divinylbenzene/Carboxen/Polydimethylsiloxane (50/30 µm coating thickness; 2 cm length; Supelco, Bellefonte, PA, USA) fiber. For this purpose, 3 g of samples was transferred into 15 mL of SPME vials (Supelco, Bellefonte, PA, USA). Then, 10 µL of two internal standards (2-methyl-3-heptanone and 2-methyl-pentanoic acid) were added into the vials. Vials

were placed on a heater at the temperature of 40 °C for 30 min to provide the accumulation of the volatiles up to headspace. After that, fiber was injected in a vial to absorb volatile compounds for 30 min. Desorption process was conducted at a temperature of 250 °C in MS sampler. A total of 3 g of the powdered almond sample was immediately placed into SPME vials (Supelco, Bellefonte, PA, USA). Then, the fiber was placed on a Shimadzu GC-2010 gas chromatography-QP-2010 mass spectrometry system (Shimadzu Corporation, Kyoto, Japan) to desorb the extracted volatiles. The separation process of extracted volatiles from the almond samples was achieved by application of DBWax column (60 m × 0.25 mm × 0.25 mm; J&W Scientific, Folsom, CA, USA). The volatile compounds were then identified according to retention index (RI) by using an n-alkane series (C10-C26) under the same conditions as mentioned above. WILEY 8 and NIST 05 mass spectral libraries were used to identify peaks.

### 2.2.6. Statistical Analysis

Statistical analysis was conducted using SPSS 22.0 software (SPSS Inc., Chicago, IL, USA). All values were recorded and expressed as mean ± standard deviation (SD). The significant differences between mean values of the almond samples were determined by analysis of variance (one way-ANOVA) using Tukey's HSD (Honestly Significant Difference) test at a significance level of $p < 0.05$. Pearson correlation coefficients were calculated with the OriginPro version 2020b (OriginLab, Northampton, MA, USA). Principal component analysis (PCA) and cluster analysis (Ward method and hierarchical) were performed using the JMP (12.2.0 SAS Institute, Inc., Cary, NC, USA).

## 3. Results

### 3.1. Some Physicochemical Properties of Almond Samples

Table 1 shows some basic physicochemical properties and total phenolic content of the almond samples. As is seen, protein content of the samples was in the range of 17.43–22.72% while the lowest protein level was for the sample of YS and the highest level was determined for the sample of NL. It was seen that the averaged protein level was 19.27%. The variation of the protein content among the samples was determined as significant ($p < 0.05$). For the oil content of the different almond samples, it was also observed that the oil levels of the samples changed significantly depending on the almond type ($p < 0.05$). It was determined that the oil levels of the almond samples ranged between 30.84–41.43%. Among all the samples, the highest oil level was determined for the sample of FS (41.43%) while the lowest oil content was monitored for the sample of LU (30.84%). Additionally, it was calculated that the average oil level of the samples was 35.37%. Ash content of the samples was also tabulated in Table 1. As is seen, the highest ash level was determined for the sample of LU (3.4%) and the lowest value was for the sample of GU (2.9%). It was observed that the difference in the ash content of the samples was found as significant ($p < 0.05$) but most of the ash values for the samples were determined as very similar to each other. It was reported that the proximate composition of almond was affected by the addition to soil, climate, and growing conditions (irrigation, fertilization, etc.) as well as differences in geographical origin, all of which have been cited as the reason for the reason of compositional change [8,10,16]. Summo et al. [2018] studied the effects of harvest time and cultivar on the chemical and nutritional characteristics of almonds, and they reported that the lipid content increased during ripening, while both protein and carbohydrate content decreased. The fatty acid composition showed a not univocal behavior during ripening and was highly influenced by the cultivar when total phenolic compounds and antioxidant activity varied among cultivars. It was also reported that the highest components found are fat and protein according to the chemical composition values of different almond varieties grown in California. It was shown that the change in composition of vitamin E and fat content and fatty acids was dependent on the harvest year, horticulture, and mainly on almond genotype [8]. Nizamlıoğlu and Nas [10] also

reported that the total oil, total ash, and protein content of the almond (Akbadem) were 52.32, 3.15 and 20.57%, respectively.

**Table 1.** Some physicochemical and bioactive properties of almond samples.

| Samples [δ] | Protein (%) | Oil (%) | Ash (%) | TPC (mg GAE/100 g) |
|---|---|---|---|---|
| **FS** | 18.85 ± 0.04 [bcd] | 41.43 ± 1.19 [a] | 3.15 ± 0.07 [ab] | 79.21 ± 1.23 [b] |
| **FL** | 18.55 ± 0.47 [cd] | 33.78 ± 1.09 [cde] | 3.3 ± 0.14 [a] | 101.03 ± 1.96 [a] |
| **MA** | 17.64 ± 0.66 [d] | 32.99 ± 0.33 [cde] | 3.25 ± 0.07 [ab] | 71.41 ± 1.17 [c] |
| **LU** | 19.61 ± 0.07 [ab] | 30.84 ± 1.34 [e] | 3.4 ± 0.14 [a] | 65.07 ± 1.5 [cd] |
| **YS** | 17.43 ± 0.34 [d] | 39.49 ± 2.57 [ab] | 3.15 ± 0.07 [ab] | 38.7 ± 1.63 [g] |
| **NL** | 22.72 ± 0.32 [a] | 33.67 ± 0.5 [cde] | 3.05 ± 0.07 [ab] | 56.41 ± 0.88 [ef] |
| **TX** | 20.11 ± 0.16 [b] | 36.92 ± 1.01 [abcd] | 3.1 ± 0.00 [ab] | 41.51 ± 1.27 [g] |
| **GU** | 17.7 ± 0.42 [d] | 37.76 ± 1.22 [abc] | 2.9 ± 0.14 [b] | 81.1 ± 1.65 [b] |
| **NU** | 20.23 ± 0.36 [b] | 32.1 ± 1.58 [de] | 3.05 ± 0.07 [ab] | 50.18 ± 0.76 [f] |
| **BA** | 19.85 ± 0.48 [ab] | 34.29 ± 1.09 [bcde] | 3.05 ± 0.07 [ab] | 62.35 ± 2.92 [de] |

[δ] Ferragnes (FS), Ferraduel (FL), Marta (MA), Lauranne (LU), Texas (TX), Nonpareil (NL), Guara (GU), Yaltinskii (YA), Nurlu (NU), and Acıbadem (BA). Different small letters in each column show the statistical significance ($p < 0.05$).

Garcia-Lopez et al. [17] reported that the lipid contents of the different almond genotypes were in the range of 53.10–61.70% for the almond samples (4 from USA, 3 from Italy, 7 from Spain, and 1 from Australia). In another study, the total lipid level of the almond samples ranged between 52.50–57.00% for four different almond genotypes from Italy [18] while the lipid concentrations of five almond genotypes (Mission, Nonpareil, Carmel, Neplus, and Peerless) from USA were in the range of 53.6–56.1%. For the Turkish almond genotypes grown in Isparta, total lipid, protein, and ash content were reported to be in the range of 44.25–55.68%, 21.23–35.27%, and 2.75–3.81%, respectively [11]. In another study, two different almond genotypes named as Akbadem and Nonpareil were compared in terms of their proximate composition and fatty acid profile. It was resulted that the total oil, ash, and protein content of both almond genotypes were 52.32 and 52.43%, 20.57 and 21.54%, and 3.15–3.26%, respectively [19]. Karaat [1] investigated the proximate composition and fatty acid profile of organic vs. conventional almonds (Ferragnes and Ferraduel) and it was revealed that the total oil, total protein, and total ash levels of organic and conventional almond were 44.5 and 46.7%, 20.9 and 20.8%, and 3.2 and 3.2%, respectively, for the Ferraduel variety. As is obviously seen from the results, the ash and protein contents were close to the reported results, but the oil content of the almond samples revealed in the current study was quite lower than that of the reported values in the literature.

Total phenolic content, which is one of the main key factors of the bioactive performance of the almond samples, was characterized and the values of total phenolic levels were also given in Table 1. As is seen from the table, it can be said that the almonds had a bioactivity and the total phenolic level ranged between 38.7–101.03 mg GAE/100 g sample. The lowest total phenolic level was observed for the sample of FL (38.7 mg GAE/100 g) and the highest total phenolic content was determined for the sample of YS (101.03 mg GAE/100 g sample). It was monitored that the differences among the sample in terms of the total phenolic levels were determined as significant ($p < 0.05$). Milbury [20] reported the total phenolic content of the different almond varieties from California were in the range of 127–241 mg GAE/100 g sample. In another study performed by Esfahlan et al. [21], total phenolic content of the almond samples from Iran ranged between 75.9–122.2 mg GAE/g for the shell and 18.1–46.6 mg GAE/g for the almond flour while the total phenolic level of the almond was 3.74 mg GAE/g in the study reported by Pinelo [22].

*3.2. Fatty Acid Profile of Almond Oil Samples*

After extraction of the oils from the almond samples, fatty acid profiles of each oil sample were determined, and the results were tabulated in Table 2. As is clearly seen from Table 2, major fatty acids detected in the almond oil samples were palmitic acid (C16:0),

palmitoleic acid (C16:1), stearic acid (C18:0), oleic acid (C18:1), and linoleic acid (C18:2). Among the fatty acids, oleic acid was determined as the main and dominant unsaturated fatty acid for the almond oils, while the palmitoleic acid was the minor unsaturated fatty acid having lower than 1% concentration in the fatty acid profile. For the palmitic acid levels of the samples, the highest level was determined for the sample of NL (8.15%), while the lowest was monitored for the sample of NU (5.59%). It was seen that there was a significant difference among the samples ($p < 0.05$). Palmitoleic acid was the minor fatty acid for all samples, with levels ranging between 0.41–0.76%. The results were similar to the palmitic acid level for the samples, and the differences were determined as significant ($p < 0.05$). Stearic acid levels also were in the range of 1.36–3.73% for the samples, and the highest stearic acid level was determined for the sample of GU and the lowest one was in the sample of TX. The level of oleic acid, which is the major and dominant fatty acid, ranged between 61.22–77.63%. The sample of NU oil, having the lowest palmitic acid and palmitoleic acid concentration, showed the highest level of oleic acid concentration (77.63%), and NL oil having the highest palmitic acid and palmitoleic acid showed the lowest oleic acid concentration (61.22%). Linoleic acid content of the almond oil samples was also given in Table 2, and the level of the linoleic acid, which is the only one polyunsaturated fatty acid in the structure of the almond oil, ranged between 13.38–27.69%. The highest linoleic acid level was for the sample of NL oil having the lowest oleic acid content and the lowest linoleic acid level was determined in the sample of FS oil. For the almond oil samples, saturated fatty acid levels (SFA), monounsaturated fatty acid levels (MUSFA), and polyunsaturated fatty acid levels (PUSFA) were calculated, and the results were also tabulated in Table 2. As is seen from the table, the lowest SFA concentration was determined for the samples of TX (7.85%) almond type, and the highest saturation level in the oil was seen in the sample of GU (10.60%). The saturation level of NL oil (10.33%) was also determined to be quite similar to the saturation level of the oil of GU almond type. Monounsaturated fatty acid (MUSFA) levels of the samples were in the range of 61.98–78.04%, while the polyunsaturated fatty acid (PUSFA) ranged between 13.38–27.69%. The highest and lowest MUSFA levels were determined for the almond oil of NU and NL samples, respectively. Thus, it can be easily said that the samples having the lowest oleic acid showed the lowest MUSFA level and vice versa. For the oil samples, PUSFA levels were equal to the levels of linoleic acid concentrations because it was the only fatty acid having double bounds in the oil samples. Thus, the PUSFA levels of the oil samples ranged between 13.38–27.69% as similar to the linoleic acid variation. Yildirim et al. [23] reported that the predominant unsaturated fatty acids of Amygdalus communis were oleic and linoleic acids, and predominant saturated fatty acid was recorded as palmitic acid. The highest oleic acid level was measured for Glorieta in both 2008 (83.35%) and 2009 (72.74%) while the highest linoleic acid content was recorded in Picantili (26.08%) in 2008 and Yaltinskii (30.01%) in 2009.

**Table 2.** Some physicochemical and bioactive properties of almond samples (%).

| Samples [δ] | PA (C16:0) | PLA (C16:1) | SA (C18:0) | OA (C18:1) | LA (C18:2) | SFA | MUSFA | PUSFA |
|---|---|---|---|---|---|---|---|---|
| FS | 6.57 ± 0.04 [ab] | 0.75 ± 0.05 [a] | 2.13 ± 0.03 [bcd] | 77.18 ± 0.75 [ab] | 13.38 ± 0.73 [b] | 8.70 | 77.93 | 13.38 |
| FL | 6.64 ± 0.06 [ab] | 0.74 ± 0.06 [a] | 1.97 ± 0.02 [cd] | 76.16 ± 1.00 [abc] | 14.51 ± 0.98 [b] | 8.60 | 76.89 | 14.50 |
| MA | 6.62 ± 0.03 [ab] | 0.61 ± 0.01 [ab] | 2.28 ± 0.04 [bc] | 71.23 ± 0.37 [bcd] | 19.28 ± 0.31 [bc] | 8.89 | 71.83 | 19.28 |
| LU | 6.93 ± 0.05 [ab] | 0.65 ± 0.01 [ab] | 2.36 ± 0.04 [b] | 69.96 ± 0.99 [d] | 20.12 ± 1.00 [abc] | 9.28 | 70.61 | 20.11 |
| YS | 7.48 ± 1.36 [b] | 0.53 ± 0.04 [bc] | 1.87 ± 0.19 [de] | 70.8 ± 2.11 [cd] | 19.33 ± 3.71 [bc] | 9.34 | 71.33 | 19.33 |
| NL | 8.15 ± 0.85 [a] | 0.76 ± 0.07 [a] | 2.18 ± 0.11 [bcd] | 61.22 ± 2.6 [e] | 27.69 ± 3.63 [a] | 10.33 | 61.98 | 27.69 |
| TX | 6.50 ± 0.11 [ab] | 0.43 ± 0.00 [c] | 1.36 ± 0.03 [f] | 69.35 ± 1.48 [d] | 22.37 ± 1.4 [ab] | 7.85 | 69.78 | 22.37 |
| GU | 6.87 ± 0.10 [ab] | 0.66 ± 0.03 [ab] | 3.73 ± 0.10 [a] | 69.6 ± 0.74 [d] | 19.15 ± 0.52 [bc] | 10.60 | 70.26 | 19.14 |
| NU | 5.59 ± 0.20 [b] | 0.41 ± 0.01 [c] | 2.42 ± 0.08 [b] | 77.63 ± 1.67 [a] | 13.96 ± 1.39 [b] | 8.00 | 78.04 | 13.96 |
| BA | 7.04 ± 0.06 [ab] | 0.51 ± 0.04 [bc] | 1.58 ± 0.08 [ef] | 72.12 ± 2.08 [abcd] | 18.76 ± 2.14 [bc] | 8.62 | 72.63 | 18.76 |

[δ] Ferragnes (FS), Ferraduel (FL), Marta (MA), Lauranne (LU), Texas (TX), Nonpareil (NL), Guara (GU), Yaltinskii (YA), Nurlu (NU), and Acıbadem (BA). PA: Palmitic acid, PLA: Palmitoleic acid, SA: Stearic acid, OA: Oleic acid, LA: Linoleic acid, SFA: Saturated fatty acid, MUSFA: Monounsaturated fatty acid, PUSFA: Polyunsaturated fatty acid. Different small letters in each column show the statistically significance ($p < 0.05$).

In a study performed by Karaat [1], the major fatty acids of two different almond genotypes (Ferragnes and Ferraduel) were reported to be oleic acid and linoleic acid. Oleic acid ranged between 78.9–82.4% and 75.8–82.4% for Ferragnes and Ferraduel almond variety, respectively, while the linoleic acid levels were in the range of 8.4–12.7% and 6.2–15.3% for the same samples, respectively. Palmitoleic acid was found to have the lowest level for all samples, as similar to the current study results. It was also informed that the oleic and linoleic acids were the most abundant unsaturated fatty acids in almonds, accounting for about 80–90%, while saturated fatty acids, such as palmitic and stearic fatty acids, are present in lower quantities (<10%) [9]. It was reported that the oleic acid and linoleic acid were reported to be the dominant fatty acids from different parts of the world such as Turkey, Spain, İtaly, Serbia, China, and California [23]. The main factor affecting the differences among the almond varieties were reported to be some growing conditions and year of cultivation. Some studies suggested that the poor water supply to the crop resulted a lower oleic/linoleic ratio indicating a significant effect of irrigation on almond fatty acid composition [24,25]. Kodad et al. [9] stated that the irrigation management and the environmental temperature levels of the growing area are the main factors affecting the total oil level and fatty acid profile of the almond varieties. They reported that the Spain almond varieties, which grow at lower temperature and abundant water supply, showed higher total oil content (58.65% vs. 55.58% ($w/w$)) and the percentage of oleic acid (71.1% vs. 68.6% ($w/w$)) compared to the ones obtained in samples grown in central Morocco. In a study conducted by Celik and Balta [25], it was reported that the USFA and SFA levels were in the range of 90.99 to 91.17% and 8.82 to 9.00%, respectively.

### 3.3. Mineral Contents of the Almond Samples

Major mineral compositions of all samples were characterized, and the results were given in Table 3. Nine different minerals (P, Mg, Ca, K, Zn, Mn, Cu, Fe, and Na) were determined as mineral elements, and it was monitored that the most abundant type of mineral was potassium (K); phosphorous (P) and magnesium (Mg) were the second and third major mineral for all almond samples. Cu and Mn were the elements showing the lowest concentration for all almond samples. Potassium levels of the almond samples were in the range of 6192.08–11,046.05 mg/kg sample. The highest K level was in the almond sample of NL and the lowest was for the sample of TX. Phosphorous (P) levels of the almond samples ranged between 4784.42–5527.03 mg/kg; the highest P levels were determined in the almond sample of FL, and the lowest P level was in the sample of YS. Mg level of the samples ranged between 2199–2657.19 mg/kg while the Ca levels of the samples were in the range of 459.34–1069.6 mg/kg. The highest Mg level was in the sample of NU (2657.19 mg/kg), while the lowest Mg was in the sample of FS. For the Ca levels, the sample of NU also showed the lowest Ca level, and for the sample of TX, Zn, Mn, Cu, Fe, and Na were the elements showing the lowest concentrations in the almond samples compared to K, P, Mg, and Ca, which, due to their levels, were trace for all almond samples. The lowest Zn level was observed in the sample of TX (33.78 mg/kg), and the highest Zn level was for the sample of FL (52.34 mg/kg). Mn levels of the samples were in the range of 12.95–24.18 mg/kg, and the sample of TX showed the highest Mn level while the sample of NU had the lowest Mn concentration (12.95 mg/kg). For the Cu levels of the samples, it was recorded that the sample of the NU showed the lowest Cu content (8.33 mg/kg) while the sample of the NL had the highest Cu concentration (15.28 mg/kg). Fe levels of the samples also ranged between 25.65–47.35 mg/kg while the levels of Na were in the range of 27.42–61.29% for all almond samples. The statistical analysis results revealed that there were significant differences among the samples in terms of the detected minerals ($p < 0.05$). Simsek and Kizmaz (2017) investigated the mineral profile of almond genotypes grown in Beyazsu district, and they reported that the potassium, phosphorous, magnesium, and calcium were in the range of 646.27–925.13 mg/100 g, 562.53–701.93 mg/100 g, 217.13–367.27 mg/100 g and 190.97–317.13 mg/100 g, respectively. It was reported that the almond kernel is accepted as a good source for the minerals [26–28]. For the almond kernels,

the major minerals were reported as K, P, Ca, and Mg and minor elements were Na, Fe, Cu, Zn, and Mn [8]. As is known, the minerals present in the plant tissues are sourced from the soil, water, and fertilizers, the differences among the genotypes in terms of mineral profiles are also attributed to the geographical origin, which combines soil and weather conditions, and agricultural practices [28]. Additionally, it was informed that the ripening stage of the kernel was also the other factor affecting the mineral profile of the almonds. Beltrán Sanahuja et al. [23] stated that different almond genotypes could maturate in periods along the year and with different ripening period length, and thus it should be considered when comparing different cultivars. Drogoudi et al. [29] investigated the mineral profile of 72 different almond genotypes provided from different countries (France, Greece, and Italy), and they found that the major elements were K, Mg, and P as similar to the current research results. Şimsek et al. [30] characterized different almond varieties including Ferraduel and Ferragnes almond types, and they reported that the major elements were also K, P, and Mg for all samples. K and P levels of the Ferraduel and Ferragnes were 903.3 and 584.7 mg/100 g and 879.4 and 621.5 mg/100 g, respectively, and it was informed as very similar to the levels found in the present work for both almond varieties. Cu was the element found in the minor levels for all almond genotypes both in the present work and the work of Simsek et al. [30].

### 3.4. Volatile Composition of the Almond Samples

Table 4 shows the volatile composition of all almond samples. For the almond samples, 41 different volatile compounds were identified, and their levels showed a significant difference among the samples. As is seen, for all almond samples, toluene, 4-octanone, pinacol, and 2-methylpentanal were the dominant volatile compounds for all almond samples. The levels of toluene ranged between 64.3–201.05 μg/kg while the highest toluene level was in the almond sample of FL and the lowest level was for the sample of YS. It was determined that the differences among the samples in terms of toluene were significant statistically ($p < 0.05$). 4-octanone levels were in the range of 29.35–334.55 μg/kg for the samples. It was observed that the highest 4-octanone concentration was monitored for the sample of FS (334.55 μg/kg) and the lowest level of 4-octanone concentration was for the sample of TX (29.35 μg/kg).

The statistical analysis revealed that the differences were significant among the almond samples in terms of 4-octanone level. Pinacol was also the most common present compound in the almond samples, and its concentration ranged between 30.45–130.1 μg/kg. The highest level of the pinacol was detected in the sample of FL, and the lowest level was recorded for the sample of TX (30.45 μg/kg). For the other abundant volatile compound named as 2-methylpentanal levels, the sample of LU showed the highest concentration (368.25 μg/kg), and the sample of YS had the lowest level of 2-methylpentanal (94.05 μg/kg). Apart from these major volatile compounds, many different constituents were detected in almond samples, and their levels were also given in Table 4. When considering all compound types, a classification was carried out and the volatile compounds were grouped as esters, alcohols, aldehydes, ketones, acids, and terpenes, and their levels were tabulated in Table 4. As is seen from Table 4, it can be said clearly that the major volatile groups were aldehydes and ketones for all almond samples.

**Table 3.** Major mineral profile of almond samples (mg/kg).

| Samples [δ] | K | Mg | Ca | P | Zn | Mn | Cu | Fe | Na |
|---|---|---|---|---|---|---|---|---|---|
| FS | 8590.31 ± 15.41 [cd] | 2199 ± 41.8 [c] | 887.35 ± 44.68 [b] | 5248.46 ± 75.27 [bc] | 52.13 ± 1.46 [a] | 19.09 ± 1.28 [bc] | 14.27 ± 0.06 [ab] | 45.99 ± 2.6 [a] | 37.42 ± 3.67 [c] |
| FL | 9196.78 ± 435.17 [bcd] | 2303.85 ± 7.18 [bc] | 883.81 ± 67.8 [b] | 5527.03 ± 77.64 [ab] | 52.34 ± 3.53 [a] | 17.37 ± 0.69 [bcd] | 14.34 ± 0.31 [ab] | 47.35 ± 0.13 [a] | 39.79 ± 1.21 [bc] |
| MA | 9788.69 ± 231.49 [abc] | 2280.6 ± 68.86 [bc] | 928.35 ± 13.64 [ab] | 4978.78 ± 94.58 [c] | 41.91 ± 3.81 [abcd] | 17.73 ± 1.32 [bcd] | 11.33 ± 0.16 [c] | 35.58 ± 0.87 [b] | 42.55 ± 0.04 [bc] |
| LU | 9775.08 ± 494.44 [abc] | 2372.97 ± 37.04 [bc] | 835.66 ± 3.97 [bc] | 5252.74 ± 184.83 [bc] | 46.16 ± 4.29 [abc] | 18.05 ± 1.03 [bc] | 11.88 ± 0.59 [c] | 41.95 ± 1.5 [ab] | 44.48 ± 4.04 [bc] |
| YS | 8956.48 ± 16.5 [cd] | 2284.41 ± 54.69 [bc] | 581.35 ± 48.45 [de] | 4784.42 ± 159.25 [c] | 39.07 ± 0.81 [cd] | 15.1 ± 0.04 [cd] | 14.9 ± 0.01 [a] | 25.65 ± 0.59 [c] | 45.92 ± 2.41 [bc] |
| NL | 11,046.05 ± 205.75 [a] | 2444.33 ± 100.94 [abc] | 719.11 ± 39.24 [cd] | 4965.88 ± 185.29 [c] | 50.04 ± 3.53 [ab] | 19.04 ± 2.11 [bc] | 15.28 ± 0.85 [a] | 35.46 ± 0.37 [b] | 43.11 ± 0.18 [bc] |
| TX | 6192.08 ± 8.39 [e] | 2251.05 ± 39.99 [bc] | 1069.6 ± 30.51 [a] | 4971.58 ± 41.13 [c] | 33.78 ± 0.23 [d] | 24.18 ± 0.04 [a] | 12.66 ± 0.31 [bc] | 36.05 ± 3.51 [b] | 41.87 ± 0.79 [bc] |
| GU | 10,364.24 ± 683.13 [ab] | 2342.57 ± 163.79 [bc] | 930.63 ± 31.93 [ab] | 5151.56 ± 80.73 [bc] | 49.56 ± 1.62 [ab] | 21.4 ± 0.34 [ab] | 13.72 ± 0.75 [ab] | 36.73 ± 1.94 [b] | 48.63 ± 3.63 [b] |
| NU | 7945.02 ± 67.71 [d] | 2657.19 ± 6.79 [a] | 459.34 ± 26.28 [e] | 5037.35 ± 72.41 [bc] | 41.45 ± 2.45 [bcd] | 12.95 ± 0.62 [d] | 8.33 ± 0.35 [d] | 46.05 ± 0.93 [a] | 61.29 ± 1.68 [a] |
| BA | 6232.97 ± 288.2 [e] | 2527.92 ± 3.42 [ab] | 662.72 ± 13.38 [d] | 5173.36 ± 313.29 [a] | 51.09 ± 0.53 [ab] | 17.23 ± 2.2 [bcd] | 14.12 ± 0.25 [ab] | 41.88 ± 0.2 [ab] | 43.8 ± 1.47 [ab] |

[δ] Ferragnes (FS), Ferraduel (FL), Marta (MA), Lauranne (LU), Texas (TX), Nonpareil (NL), Guara (GU), Yaltinskii (YA), Nurlu (NU), and Acıbadem (BA). Different small letters in each column show the statistically significance ($p < 0.05$).

**Table 4.** Volatile profile of almond samples [δ].

| No. | Compound | RI | FS | FL | MA | LU | YS | NL | TX | GU | NU | BA |
|---|---|---|---|---|---|---|---|---|---|---|---|---|
| 1 | Butanal | 833 | 6.4 ± 0.42 [a] | 3.15 ± 0.35 [b] | n.d | 2.1 ± 0.42 [ab] | 0.6 ± 0.28 [de] | n.d | n.d | 1.1 ± 0.57 [cde] | 1.3 ± 0.28 [bcd] | 1.85 ± 0.35 [abc] |
| 2 | 2-Methylbutanal | 894 | n.d | n.d | n.d | n.d | n.d | 0.45 ± 0.07 [a] | 0.3 ± 0.14 [a] | n.d | n.d | 0.35 ± 0.07 [a] |
| 3 | 3-Methylbutanal | 920 | n.d | n.d | n.d | n.d | n.d | 0.15 ± 0.07 [b] | n.d | n.d | 0.15 ± 0.07 [b] | 0.45 ± 0.21 [a] |
| 4 | Ethanol | 942 | n.d | n.d | n.d | n.d | n.d | 3.45 ± 0.92 [a] | 2.15 ± 0.49 [ab] | n.d | 0.9 ± 0.28 [bc] | n.d |
| 5 | 3-Methyl-2-pentanone | 1023 | n.d | 0.4 ± 0.14 [a] | 0.65 ± 0.07 [a] | n.d | 0.25 ± 0.07 [a] | n.d | n.d | n.d | n.d | n.d |
| 6 | α-pinene | 1025 | n.d | n.d | n.d | n.d | 0.55 ± 0.2 [bc] | 1.05 ± 0.21 [b] | 2.15 ± 0.35 [a] | 0.25 ± 0.07 [c] | 1.1 ± 0.28 [b] | n.d |
| 7 | Toluene | 1044 | 161.45 ± 9.55 [b] | 201.05 ± 7.57 [a] | 103.1 ± 7.92 [c] | 145.6 ± 8.2 [b] | 64.3 ± 5.52 [f] | 85.9 ± 6.22 [cdef] | 69.1 ± 2.4 [ef] | 95.1 ± 7.5 [cde] | 75.3 ± 5.37 [def] | 101.4 ± 6.65 [cd] |
| 8 | Butyl acetate | 1070 | 74.85 ± 3.18 [a] | 87.05 ± 6.29 [a] | 18.95 ± 1.63 [def] | 57.2 ± 6.08 [b] | 27.7 ± 2.26 [de] | 15.9 ± 3.68 [ef] | 11.6 ± 2.55 [f] | 26.65 ± 3.89 [def] | 45.85 ± 3.75 [bc] | 31.9 ± 1.7 [cd] |
| 9 | Hexanal | 1084 | n.d | n.d | 10.6 ± 1.13 [c] | n.d | n.d | 18.05 ± 2.33 [b] | 23.2 ± 2.26 [a] | 8.3 ± 1.27 [c] | n.d | 11.05 ± 0.35 [c] |
| 10 | 2-Methyl-1-propanol | 1088 | 6.4 ± 0.71 [a] | 6.9 ± 0.85 [a] | n.d | 2.15 ± 0.35 [b] | n.d | n.d | 1.35 ± 0.21 [bc] | n.d | 0.55 ± 0.21 [c] | n.d |
| 11 | 6-hepten-3-one-4-methyl | 1115 | 3.2 ± 0.14 [b] | n.d | n.d | n.d | 1.55 ± 0.35 [d] | n.d | 4.7 ± 0.28 [a] | n.d | n.d | 2.45 ± 0.35 [c] |
| 12 | isoamyl acetate | 1122 | n.d | 1.6 ± 0.28 [a] | 1.15 ± 0.07 [b] | n.d | n.d | n.d | n.d | n.d | n.d | n.d |
| 13 | Ethylbenzene | 1126 | 23.45 ± 1.06 [b] | 22.1 ± 1.7 [bc] | 17.6 ± 1.56 [cd] | 28.9 ± 1.7 [a] | 23.4 ± 1.27 [b] | n.d | n.d | 10.25 ± 0.64 [e] | 17.15 ± 1.48 [d] | 8.6 ± 1.13 [e] |
| 14 | 2.5-Dimethyl-3-hexanone | 1144 | 24.55 ± 2.47 [a] | 26.55 ± 1.63 [a] | n.d | n.d | n.d | 16.1 ± 2.4 [b] | 10.85 ± 0.49 [c] | n.d | 6.55 ± 0.21 [c] | 27.2 ± 0.57 [a] |
| 15 | butyl isobutyrate | 1150 | n.d | 2.15 ± 0.49 [a] | n.d | 1.15 ± 0.35 [b] | n.d | n.d | n.d | n.d | n.d | 0.650.21 [bc] |
| 16 | butanol | 1155 | 12.4 ± 0.99 [cd] | 14.65 ± 0.64 [bc] | 7.9 ± 0.57 [ef] | 17.8 ± 1.56 [b] | 5.45 ± 0.92 [f] | 22.6 ± 1.13 [a] | 15.45 ± 0.92 [bc] | 10.05 ± 0.64 [de] | 12.55 ± 0.92 [cd] | 8.9 ± 0.42 [def] |
| 17 | 3-heptanone | 1158 | 7.35 ± 0.64 [b] | 9.25 ± 0.78 [a] | n.d | 10.05 ± 0.92 [a] | n.d | n.d | n.d | 3.3 ± 0.28 [c] | n.d | 2.6 ± 0.28 [c] |
| 18 | 2-heptanone | 1184 | n.d | n.d | n.d | n.d | 2.1 ± 0.28 [b] | n.d | 1.6 ± 0.14 [b] | n.d | 3.05 ± 0.35 [a] | n.d |
| 19 | Limonene | 1195 | n.d | n.d | 3.85 ± 0.49 [a] | n.d | n.d | 1.65 ± 0.21 [c] | n.d | 2.6 ± 0.28 [b] | 0.75 ± 0.21 | n.d |
| 20 | 3-Methyl-butanol | 1195 | 1.05 ± 0.35 [c] | n.d | n.d | 1.7 ± 0.14 [c] | 5.4 ± 0.42 [a] | n.d | 3.9 ± 0.28 [b] | n.d | n.d | n.d |
| 21 | butyl butyrate | 1206 | n.d | 1.4 ± 0.28 [b] | n.d | 2.25 ± 0.21 [a] | n.d | n.d | n.d | n.d | n.d | n.d |
| 22 | 2-hexanol | 1220 | 1.6 ± 0.28 [c] | n.d | 2.25 ± 0.21 [ab] | n.d | 0.7 ± 0.14 [d] | n.d | 2.75 ± 0.21 [a] | 0.8 ± 0.14 [d] | 1.45 ± 0.21 [c] | 1.85 ± 0.07 [bc] |
| 23 | 4-octanone | 1230 | 334.55 ± 8.27 [a] | 318.5 ± 6.79 [a] | 56.7 ± 3.68 [de] | 129.8 ± 7.5 [c] | 143.65 ± 5.44 [c] | 40.45 ± 2.62 [ef] | 29.35 ± 1.2 [f] | 75.25 ± 5.59 [d] | 83.45 ± 8.7 [d] | 172.65 ± 12.09 [b] |
| 24 | 3-octanone | 1254 | 1.7 ± 0.28 [a] | 2.05 ± 0.35 [a] | n.d | n.d | n.d | n.d | 0.75 ± 0.21 [b] | 0.55 ± 0.21 [bc] | n.d | n.d |
| 25 | m-Cymene | 1262 | n.d | 2.75 ± 0.49 [b] | n.d | 4.55 ± 0.21 [a] | 0.5 ± 0.14 [cd] | n.d | 1.05 ± 0.21 [c] | n.d | n.d | n.d |

**Table 4.** *Cont.*

| No. | Compound | RI | FS | FL | MA | LU | YS | NL | TX | GU | NU | BA |
|---|---|---|---|---|---|---|---|---|---|---|---|---|
| 26 | butyl valerate | 1299 | 1.05 ± 0.21 [a] | 0.85 ± 0.35 [a] | n.d | n.d | 0.65 ± 0.21 [a] | n.d | n.d | n.d | n.d | n.d |
| 27 | 2-heptenal | 1332 | n.d | n.d | 2.4 ± 0.28 [a] | n.d | n.d | 1.6 ± 0.28 [bc] | n.d | 2.4 ± 0.14 [a] | 1.2 ± 0.14 [c] | 1.85 ± 0.21 [ab] |
| 28 | Pinacol | 1336 | 120.65 ± 11.38 [ab] | 130.1 ± 8.91 [a] | 40.2 ± 2.69 [g] | 87.1 ± 5.09 [cd] | 73.4 ± 3.25 [de] | 62.75 ± 2.05 [ef] | 30.45 ± 0.92 [g] | 50.55 ± 1.06 [fg] | 103.15 ± 4.17 [bc] | 40.6 ± 1.13 [g] |
| 29 | 1-hexanol | 1357 | 0.3 ± 0.14 [b] | n.d | n.d | n.d | 0.25 ± 0.07 [b] | n.d | 1.05 ± 0.35 [a] | n.d | n.d | n.d |
| 30 | 2-nonanone | 1396 | 11.45 ± 0.78 [bc] | 14.1 ± 1.84 [b] | 8.15 ± 0.35 [cde] | 21.65 ± 2.05 [a] | n.d | 7.05 ± 0.35 [de] | 12.85 ± 0.64 [b] | 10.85 ± 0.64 [bc] | 4.7 ± 0.57 [e] | 13.55 ± 0.78 [b] |
| 31 | 2-methylpentanal | 1451 | 296.55 ± 15.91 [bc] | 327.6 ± 9.62 [ab] | 175.8 ± 10.75 [cd] | 368.25 ± 16.76 [a] | 94.05 ± 6.15 [f] | 131.15 ± 4.74 [f] | 110.9 ± 10.32 [f] | 202.05 ± 7.71 [c] | 136.75 ± 4.74 [de] | 262.8 ± 16.12 [c] |
| 32 | Benzaldehyde | 1541 | n.d | n.d | 4.7 ± 0.28 [d] | 1.55 ± 0.35 [ef] | 3.25 ± 0.49 [de] | 15.6 ± 0.71 [a] | 12.55 ± 0.92 [b] | 6.8 ± 0.42 [c] | n.d | 7.15 ± 0.64 [c] |
| 33 | Isobutyric acid | 1577 | 35.45 ± 3.32 [a] | 37.6 ± 3.11 | n.d | 11.75 ± 2.33 [b] | n.d | n.d | n.d | n.d | n.d | n.d |
| 34 | γ-butyrolactone | 1625 | n.d | n.d | 0.35 ± 0.07 [a] | n.d | n.d | n.d | n.d | n.d | 0.4 ± 0.14 [a] | n.d |
| 35 | 2-Decenal | 1646 | n.d | n.d | 0.8 ± 0.14 [bc] | n.d | 0.45 ± 0.07 [cd] | 1.1 ± 0.14 [ab] | 1.55 ± 0.21 [a] | 1.25 ± 0.21 [ab] | n.d | 1.3 ± 0.14 [a] |
| 36 | Phenylacetaldehyde | 1668 | n.d | n.d | 4.05 ± 0.49 [b] | 1.15 ± 0.21 [c] | n.d | 5.8 ± 0.57 [a] | n.d | 4.75 ± 0.64 [ab] | n.d | n.d |
| 37 | valeric acid | 1719 | 2.05 ± 0.21 [a] | 1.1 ± 0.28 [bc] | n.d | n.d | 1.45 ± 0.21 [ab] | n.d | 1.05 ± 0.35 [bc] | n.d | 1.05 ± 0.21 [bc] | 0.55 ± 0.07 [cd] |
| 38 | 2-Phenyl-2-propanol | 1768 | 10.15 ± 0.64 [b] | 8.65 ± 0.64 [ab] | 2.75 ± 0.21 [e] | 13.7 ± 1.13 [a] | 2.2 ± 0.42 [e] | 3.6 ± 0.28 [de] | 7.05 ± 0.49 [bc] | 4.2 ± 0.42 [de] | 5.35 ± 0.64 [cd] | 8.45 ± 0.92 [ab] |
| 39 | γ-decalactone | 2185 | 11.1 ± 0.85 [b] | n.d | 15.55 ± 1.2 [a] | n.d | n.d | n.d | n.d | 7.9 ± 0.42 [c] | 1.65 ± 0.21 [de] | 2.65 ± 0.35 [d] |
| 40 | γ-undecalactone | 2246 | 2.85 ± 0.49 [b] | 5.2 ± 0.42 [a] | n.d | 2.6 ± 0.28 [b] | 0.3 ± 0.14 [c] | n.d | 0.5 ± 0.14 [c] | n.d | n.d | n.d |
| 41 | Diethyl Phthalate | 2374 | 0.35 ± 0.07 [bc] | 5.75 ± 0.64 [a] | 1.1 ± 0.42 [b] | n.d | n.d | 1.05 ± 0.21 [b] | n.d | n.d | 1.05 ± 0.21 [b] | 0.65 ± 0.07 [bc] |
| | Total (µg/kg) | | | | | | | | | | | |
| | Esters | | 76.25 | 98.8 | 21.2 | 60.6 | 28.35 | 16.95 | 11.6 | 26.65 | 47.55 | 32.55 |
| | Alcohols | | 152.55 | 160.3 | 53.1 | 122.45 | 87.4 | 92.4 | 64.15 | 65.6 | 123.95 | 59.8 |
| | Aldehydes | | 302.95 | 330.75 | 198.35 | 373.05 | 98.35 | 173.9 | 148.5 | 226.65 | 139.4 | 286.8 |
| | Ketones | | 396.75 | 376.05 | 81.4 | 164.1 | 147.85 | 63.6 | 60.6 | 97.85 | 99.8 | 221.1 |
| | Acids | | 37.5 | 38.7 | 0 | 11.75 | 1.45 | 0 | 1.05 | 0 | 1.05 | 0.55 |
| | Terpenes | | 0 | 2.75 | 3.85 | 4.55 | 1.05 | 2.7 | 3.2 | 2.85 | 1.85 | 0 |
| | Others | | 184.9 | 223.15 | 120.7 | 174.5 | 87.7 | 85.9 | 69.1 | 105.33 | 92.45 | 110 |

[δ] Ferragnes (FS), Ferraduel (FL), Marta (MA), Lauranne (LU), Texas (TX), Nonpareil (NL), Guara (GU), Yaltinskii (YA), Nurlu (NU), and Acıbadem (BA). Different small letters in each column show the statistically significance ($p < 0.05$); n.d: not determined.

The minor chemical volatile group was also monitored as acids and terpenes (Table 4). The highest ester levels were determined in the sample of FL (98.8 μg/kg) while the lowest ester levels were recorded for the almond sample of TX (11.6 μg/kg). Alcohols ranged between 53.1–160.3 μg/kg while the sample of FL also showed the highest alcohol group and the sample of MA showed the lowest alcohol group levels. The almond samples of LU and FS showed the highest levels of aldehydes and ketones, respectively, while the samples of YS and TX had the lowest levels of aldehydes and ketones, respectively. For all almond samples, other compounds that do not place in the mentioned chemical class above were also characterized, and their levels were also tabulated in Table 4. For all identified volatile compounds, almond types showed a significant effect on the variation among the sample in terms of volatile compound concentration.

Xiao et al. [31] characterized volatiles in raw and dry-roasted almonds (*Prunus dulcis*); using NIST libraries and Kovats Index values, they identified 41 volatiles including 13 carbonyls, 1 pyrazine, 20 alcohols, and 7 additional volatiles. It was revealed that predominant volatile (2934.6 ± 272.5 ng/g) present in raw almonds was benzaldehyde, known as the breakdown product of amygdalin. In the present study, benzaldehyde was not the dominant compound, and it was detected in the almond samples except FS, FL, and NU while the sample of NL showed the highest benzaldehyde concentration. Mexis et al. [32] also reported the similar results to the results of Xiao et al. [31].

Karaat [1] investigated the volatile profile of two almond genotypes, and he reported that the detected volatile compounds were butanal, butyl acetate, ethylbenzene, p-chlorotoluene, and 4-octanone. In another study, Erten and Cadwallader [33] reported the identification of predominant aroma components of raw, dry roasted, and oil roasted almonds, and they revealed that the 1-octen-3-one and acetic acid were important aroma compounds in raw almonds. In most of the study, the authors reported the volatile profile of raw almonds revealed that the octanal, nonanal, acetic acid, and methional were the volatile constituents of raw almonds [31,34–36].

### 3.5. Principal Component and Correlation Analysis

To determine the relationship among the studied parameters, Pearson correlation was performed, and the results were illustrated in Figure 1. As is seen, there were generally no significant correlations among the proximate composition except fatty acid and mineral profile. It was determined that a significant and negative and positive correlation was observed between oleic acid and palmitoleic acid ($r = -0.81$) and linoleic acid and palmitic acid ($r = 0.74$), respectively. Additionally, for all almond varieties, the sample having the highest oleic acid content had the lowest linoleic acid level, and thus a negative correlation between oleic and linoleic acid level was monitored ($r = -0.98$). In addition to that, between the palmitoleic acid, zinc, and phosphorus contents and total phenolic levels of the almond samples, it was determined that there was a positive correlation. For the minerals, between Fe and P ($r = 0.82$), Na and Mg (0.78), Mn and Ca (0.87), and Mn and K (0.87), significant positive correlations were observed while Ca and Mg and K and Mg showed negative correlation.

For the volatile compounds, principal component (PC) analysis was performed and two main PCs explaining the total variation of the parameters as 55.2% was determined. Figure 2 top shows the factor loadings of the samples and volatile compounds, and it was determined that the almond samples showed differences in terms of the aroma. As is seen from the figure, TX, NL, BA, and YS were in the same class and showed significantly different volatile compounds compared to FL and LU. Additionally, FS differed in terms of aroma profile compared to MA, NU, and GU. BA from Turkey showed similar aroma with the sample of TX and NL from USA. Figure 2 bottom also showed the classification of the identified volatile compounds. As is seen, five main clusters were determined for the volatile profile of the samples, and it was also observed that the major compounds identified for all of the studied almond genotypes were aldehydes, ketones, and alcohols.

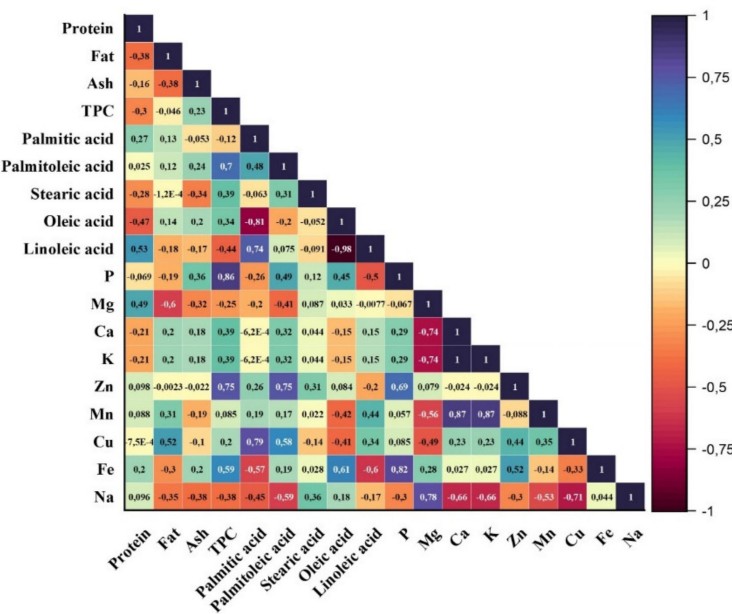

**Figure 1.** Pearson correlation matrix of almond characteristics.

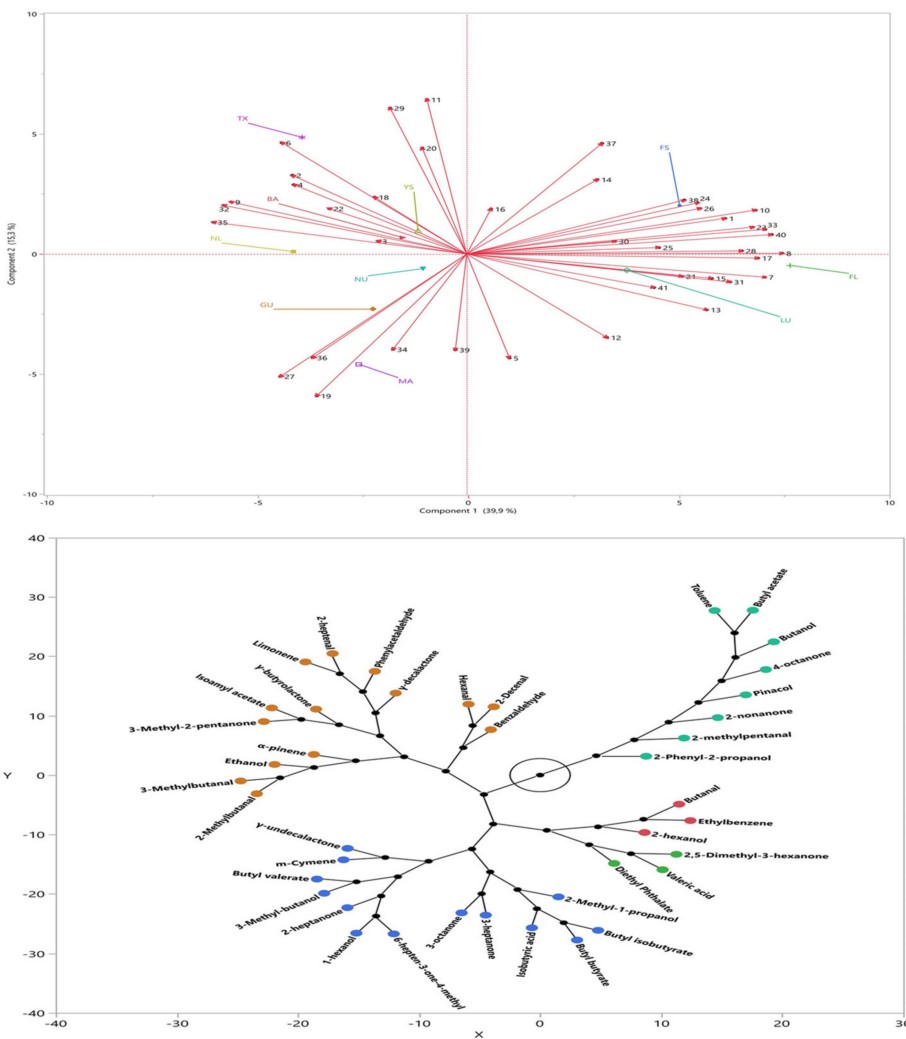

**Figure 2.** Factor loadings (**top**) and cluster (**bottom**) of the volatile compounds for almond samples.

### 3.6. Conclusions

It was observed that different almond genotypes from different countries showed a significant variation in terms of main proximate composition, fatty acid and mineral profile, and volatile composition. All almond samples were determined as rich in fat and protein. Additionally, they had phenolic compounds. It was revealed that the almond oils were composed of mainly unsaturated fatty acids such as oleic acid. For the mineral profile of the samples, all of them showed richness in terms of phosphorus, potassium, and magnesium. After volatile characterization, especially, aldehydes, ketones, and alcohols were the major dominant aroma compounds, and the almond varieties differed significantly in terms of the identified volatile component. It was concluded that the almonds showed significant difference according to the genotypes as well as the origin. This comparative research presented the main proximate composition and values compounds (fatty acid, mineral, and aroma) concentrations of the different almond genotypes to see the differences in the nutritive values of the studied varieties.

**Funding:** This research received no external funding.

**Institutional Review Board Statement:** Not applicable.

**Informed Consent Statement:** Not applicable.

**Data Availability Statement:** Not applicable.

**Acknowledgments:** I would like to thank Fırat EGE (Adıyaman University, Faculty of Agriculture, and Department of Horticulture) for the help to provide the almond samples.

**Conflicts of Interest:** The author declares no conflict of interest.

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
