# Peer review of "A Detailed Comparative Study on Some Physicochemical Properties, Volatile Composition, Fatty Acid, and Mineral Profile of Different Almond (Prunus dulcis L.) Varieties"

_horticulturae, doi:10.3390/horticulturae8060488_

Round 1

Reviewer 1 Report

The manuscript, A detailed comparative study on some physicochemical properties, aromatic composition, fatty acid, and mineral profile of  different almond (Prunus dulcis L.) varieties, presents original research which appears to be based on sound scientific methods and provides important information with both basic and applied science applications. Extensive editing of English language style will be required prior to publication though required edits, while numerous, are all relatively minor and the author's intent and presentations appear clear and unambiguous throughout.

The term 'aromatic' used in the title and throughout can refer to either 1) aroma or smell or 2) a type of chemical configuration in certain organic molecules (including many listed in Table 4). Consequently, the authors should either define the usage or use an alternate term in the manuscript.

The variety of Marta is a Spanish variety not French.

It would be useful to have information on average kernel mass as well as relative time of maturity (harvest) for each of the evaluated varieties to help in the analysis of the data.

Similarly, as the authors have pointed out that growing environment can affect kernel properties, it would be useful to have a brief summary of the growth environment for the samples.

Line 202. [The highest total phenolic level was observed for the sample of FL (38.7 mg GAE/100 g) and the highest total phenolic content was determined for the sample of YS (101.03 mg GAE/100 204 g sample). ] Does highest total phenolic level actually refer to lowest total phenolic content?].

It should be pointed out in the introduction that Amygdalus communis is a synonym for Prunus dulcis.

The information on volatile composition was very interesting. Besides benzaldehyde, are there any reports identifying the importance of individual volatiles to final sensory or eating quality? This information should be included if available.

Finally, the authors are to be commended for their selection of varieties for testing as these varieties represent important parents in the breeding programs worldwide. In this context, it would be useful also to know the parentage of the Turkish varieties Nurlu and Acıbadem. [Yaltinski is also spelled  Yaltinskii and is from Yalta, Crimea, within the former Soviet Union].

Author Response

REVİEWERS

The manuscript, A detailed comparative study on some physicochemical properties, aromatic composition, fatty acid, and mineral profile of different almond (Prunus dulcis L.) varieties, presents original research which appears to be based on sound scientific methods and provides important information with both basic and applied science applications. Extensive editing of English language style will be required prior to publication though required edits, while numerous, are all relatively minor and the author's intent and presentations appear clear and unambiguous throughout.

The term 'aromatic' used in the title and throughout can refer to either 1) aroma or smell or 2) a type of chemical configuration in certain organic molecules (including many listed in Table 4). Consequently, the authors should either define the usage or use an alternate term in the manuscript.

Reply: It is revised as “volatile”

The variety of Marta is a Spanish variety not French.

Reply: It is corrected as suggested.

It would be useful to have information on average kernel mass as well as relative time of maturity (harvest) for each of the evaluated varieties to help in the analysis of the data.

Reply: The requested information is added in the proper place in the manuscript.

Similarly, as the authors have pointed out that growing environment can affect kernel properties, it would be useful to have a brief summary of the growth environment for the samples.

Reply: The requested information is added in the proper place in the manuscript.

Line 202. [The highest total phenolic level was observed for the sample of FL (38.7 mg GAE/100 g) and the highest total phenolic content was determined for the sample of YS (101.03 mg GAE/100 204 g sample). ] Does highest total phenolic level actually refer to lowest total phenolic content?].

Reply: It is corrected as suggested.

It should be pointed out in the introduction that Amygdalus communis is a synonym for Prunus dulcis.

Reply: It is revised as suggested.

The information on volatile composition was very interesting. Besides benzaldehyde, are there any reports identifying the importance of individual volatiles to final sensory or eating quality? This information should be included if available.

Reply: According to my literature survey, there was no study reporting the identified individual volatiles affected final sensory or eating quality.

Finally, the authors are to be commended for their selection of varieties for testing as these varieties represent important parents in the breeding programs worldwide. In this context, it would be useful also to know the parentage of the Turkish varieties Nurlu and Acıbadem. [Yaltinski is also spelled  Yaltinskii and is from Yalta, Crimea, within the former Soviet Union].

Reply: The correction is performed.

Author Response

REVİEWER 2

Review of manuscript number Horticulture - 1720000

Title: A detailed comparative study on some physicochemical properties aromatic composition, fatty acid, and mineral profile of different almond (Prunus dulcis L.)varieties

1) The article describes the results of the investigation on oil, protein, ash, fatty acids, total phenolic content, minerals, and aromatic profiles in ten different almond genotypes. The paper needs English language revision.

Reply: The manuscript is double checked for the grammatical problems as requested.

2) The abstract does not present the conclusion of the paper.

Reply: It is revised as suggested.

Introduction

3) The introduction is rather short and the aim of the study is not clear. After characterization of the composition of the samples and performing cluster analysis, there is no specific aim of the study given.

Reply: The main aim of the study was basically comparison for the different origin of almond genotypes.  This is the only aim of the study.

Material and methods

4) Can the authors explain in which country, the region the plants were harvested?

Reply: The different almond genotypes were cultivated in Adıyaman and the samples were provided by the Agricultural Credit Cooperative Adıyaman Kahta Almond-Pistachio Enterprise in harvest year of 2021.

5) Looking at the oil content results I think that usage of n-hexane during Soxhlet extraction was not adequate, because your results are rather lower than they should be about 50% or even higher.

Reply: According to the literature, n nexane is quite effective solvent for the extraction of the oil from the food or seed samples. In a study performed by Egenethan et al. (2006) [Eganathan, P., Subramanian, H. S., Latha, R., & Rao, C. S. (2006). Oil analysis in seeds of Salicornia brachiata. Industrial Crops and Products, 23(2), 177-179.] petroleum ether and n-hexane were compared the total oil content of the samples were determined to be quite higher when n-hexane was used as a extractşon solvent for the sample of Salicornia brachiate compared to petroleum ether. So, I don’t think that the lower oil levels caused by the using of n-hexane in Soxhlet extraction system. In addition to that, When the article of Karaat (2019) [Karaat, F. E. (2019). Organic vs conventional almond: market quality, fatty acid composition and volatile aroma compounds. Applied Ecology and Environmental Research, 17(4), 7783-7793] is examined, we can say that the total oil content of Ferraduel and Ferragnes various almond samples is close to our results. Also, it is observed that there are significant oil content differences even in organic or conventional cultivation of the same type of almond samples. In another research of Karaat (2020) [Karaat, F. E. (2020). A comparative study on pomological traits, fatty acid composition and volatile aroma compounds of irrigated and rain-fed almond. Acta Scientiarum Polonorum-Hortorum Cultus, 19(1), 141-149], Again, when the article was examined, significant differences were found in the oil content of the almond samples produced with and without water, and it was observed that the total oil content was lower than the results we found.

Page 3 line 91

6) Can the author explain why you added 46 ml of water 100 ul of sample and 1 ml of ethanol?and then you added 1 ml of Folin Ciocalteu’s reagent. Should those samples have been defatted? What is the information about the calibration curve?

Reply: To determine the total phenolic content of the samples, the Folin Ciocalteu’s reagent method was followed as suggested by Yıldırım et al. [2001] and Durmaz and Alpaslan (2007).

  • Yıldırım, A.; Mavi, A.; Kara, A.A. Determination of Antioxidant and Antimicrobial Activities of Rumex crispus L. Extracts. J Agric Food Chem. 2001, 49(8), 4083-4089.
  • Durmaz, G.; Alpaslan, M. Antioxidant Properties of Roasted Apricot (Prunus armeniaca L.) kernel. Food Chem. 2007, 100(3), 1177-1181.

7) Page 3 line 123 How authors identified the fatty acids?

Reply: Fatty acid composition was evaluated by determination of palmitic acid, palmitoleic acid, stearic acid, oleic acid, linoleic acid contents. The fatty acid methyl esters were detected in a gas chromatograph (Shimatzu, QP2010 ULTRA) with a flame ionization detector and Rtx-5 MS capillary column according to Anonymous (1993). Obtained results were expressed as percentages of each fatty acid with regard to total oil content.

Anonymous (1993): Me´todos oficiales de ana´lisis; preparacio´n de los e´steres metı´licos (Official methods of analysis; preparation of metallic esters). – Ministerio de Agricultura Pesca y Alimentacio´n; Direccio´n General de Polı´tica Alimentaria (Ministry of Agriculture, Fisheries and Food; General Directorate of Food Policy); Madrid/Spain,41(2): 135.

8) Page 4 line 140 please change desorption process was performed in GC sampler, not MS sampler. The temperature program was not given.

Reply: It is revised as suggested.

Results and discussion

9) The biggest disadvantage of the paper is that the authors do not see the point in the comparison of the given results between analyzed genotypes of almonds. In some cases for example low content of fat, the authors do not explain why their results were lower than those of other scientists – page 5 line 198.

Reply: When the article of Karaat (2019) [Karaat, F. E. (2019). Organic vs conventional almond: market quality, fatty acid composition and volatile aroma compounds. Applied Ecology and Environmental Research, 17(4), 7783-7793] is examined, we can say that the total oil content of Ferraduel and Ferragnes various almond samples is close to our results. Also, it is observed that there are significant oil content differences even in organic or conventional cultivation of the same type of almond samples. In another research of Karaat (2020) [Karaat, F. E. (2020). A comparative study on pomological traits, fatty acid composition and volatile aroma compounds of irrigated and rain-fed almond. Acta Scientiarum Polonorum-Hortorum Cultus, 19(1), 141-149], Again, when the article was examined, significant differences were found in the oil content of the almond samples produced with and without water, and it was observed that the total oil content was lower than the results we found.

10) Can you describe why the TPC content was so varied in the different genotypes of almonds? can you give some examples?

Reply: The genotype and climatic conditions generally effect the proximate composition and bioactive properties of the cultivated agricultural products. In the section of the total phenolics in the manuscript, different literature survey results are given such as:

Milbury et al. (2006) reported the total phenolic content of the different almond varieties from California were in the range of 127-241 mg GAE/100 g sample. In another study per-formed by Esfahlan et al. [2010], total phenolic content of the almond samples from Iran ranged between 75.9-122.2 mg GAE/g for the shell and 18.1-46.6 mg GAE/g for the al-mond flour while the total phenolic level of the almond was 3.74 mg GAE/g in the study reported by Pinelo et al. (2004).

  • Milbury, P.E.; Chen, C.Y.; Dolnikowski, G.G.; Blumberg, J.B.; Lumberg, J.E.B.B. Determination of Flavonoids and Phenolics and Their Distribution in Almonds. J. Agric. Food Chem. 2006, 54, 5027–5033
  • Esfahlan, A.J.; Jamei, R.; Esfahlan, R.J. The importance of almond (Prunus amygdalus L.) and its by-products. Food Chem. 2010, 120, 349–360.
  • Pinelo, M.; Rubilar, M.; Sinero, J.; Nunez, M.J. Extraction of antioxidant phenolics from almond hulls (Prunus amygdalus) and pine sawdust (Pinus pinaster). Food Chem. 2004, 85(2), 267-273.

As is seen, there is a wide variation for the total phenolic content of the almond samples as also reported following paper.

  • Summo, C., Palasciano, M., De Angelis, D., Paradiso, V. M., Caponio, F., & Pasqualone, A. (2018). Evaluation of the chemical and nutritional characteristics of almonds (Prunus dulcis (Mill). DA Webb) as influenced by harvest time and cultivar. Journal of the Science of Food and Agriculture, 98 (15), 5647-5655.

11) The most abundant aroma compounds of almonds presented in the paper for instance tolüene and pinacol are rather impossible to be found in such amounts in this kind of sample, I think that samples were contaminated in the laboratory, or during sample preparation in another step, because other authors do not present that toluene is found such high amounts. Pinacol was not found in such samples. Why authors do not explain such observations? Statistical analysis of results is rather skimpy and does not explain all of the relationships between obtained results. What was the purpose of this part of the article? I do not see the main result of the statistical analysis.

Reply: In both articles below, toluene compounds were detected at a significant level.

Vázquez-Araújo, L., Enguix, L., Verdú, A., García-García, E., & Carbonell-Barrachina, A. A. (2008). Investigation of aromatic compounds in toasted almonds used for the manufacture of turrón. European Food Research and Technology, 227(1), 243-254.

Agila, A., & Barringer, S. (2012). Effect of roasting conditions on color and volatile profile including HMF level in sweet almonds (Prunus dulcis). Journal of food science, 77(4), C461-C468.

In both articles below, toluene compounds were detected at a significant level.

Karaat, F. E. (2019). Organic vs conventional almond: market quality, fatty acid composition and volatile aroma compounds. Applied Ecology and Environmental Research, 17(4), 7783-7793.

Karaat, F. E. (2020). A comparative study on pomological traits, fatty acid composition and volatile aroma compounds of irrigated and rain-fed almond. Acta Scientiarum Polonorum-Hortorum Cultus, 19(1), 141-149.

Conclusion:

12) The section is not clear and cannot understand what the main conclusion is, before reading this article I did know that the composition of the almond samples would be different but how can you explain which genotype was the best?

Reply: This study aimed to make a basic comparison for some almond genotypes in terms of their main proximate composition, fatty acid profile, mineral and volatile composition. So the reader could decide which genotype is best because one genotype having good fatty acid profile showed quite week phenolic content. So, there are big variations among the studied sample.

Reviewer 3 Report

Manuscript ID: horticulturae- 1720000

Title: A detailed comparative study on some physicochemical properties, aromatic composition, fatty acid, and mineral profile of different almond (Prunus dulcis L.) varieties

Authors: Okan LEVENT

 Review of the manuscript

Comments

The manuscript presented for revision is interesting, but the results and conclusions are not very original. Similar works and studies have been reported in recent years (e.g.  Kodad, O., Socias, and Company, R., & Alonso, J. M. (2018). Genotypic and environmental effects on tocopherol content in almond. Antioxidants, 7 (1), 6;

Özcan, M. M., Matthäus, B., Aljuhaimi, F., Mohamed Ahmed, I. A., Ghafoor, K., Babiker, E. E., ... & Alqah, H. A. (2020). Effect of almond genotypes on fatty acid composition, tocopherols and mineral contents and bioactive properties of sweet almond (Prunus amygdalus Batsch spp. Dulce) kernel and oils. Journal of Food Science and Technology, 57 (11), 4182-4192;

Ibourki, M., Bouzid, H. A., Bijla, L., Aissa, R., Ainane, T., Gharby, S., & El Hammadi, A. (2022). Physical fruit traits, proximate composition, fatty acid and elemental profiling of almond [Prunus dulcis Mill. DA Webb] kernels from ten genotypes grown in southern Morocco. OCL, 29, 9;

Simsek, M., Gulsoy, E., Yavic, A., Arikan, B., Yildirim, Y., Olmez, N., ... & Boguc, F. (2018). Fatty acid, mineral and proximate compositions of various genotypes and commercial cultivars of sweet almond from the same ecological conditions. Appl. Ecol. Environ. Res, 16 (3), 2957-71.

I have a few comments about the manuscript:

 Introduction:

Line 28. Please compare the information on the volume of almond production in recent years with the FAOSTAT data.

Line 31-41: When discussing the nutritional and health benefits of almonds, please refer to the new articles e.g. Kalita, S., Khandelwal, S., Madan, J., Pandya, H., Sesikeran, B., & Krishnaswamy, K. (2018). Almonds and cardiovascular health: A review. Nutrients, 10 (4), 468.

Mathpal, D., & Rathore, G. (2021). An analysis of health benefits of almonds. ACADEMICIA: An International Multidisciplinary Research Journal, 11 (11), 903-910.

Summo, C., Palasciano, M., De Angelis, D., Paradiso, V. M., Caponio, F., & Pasqualone, A. (2018). Evaluation of the chemical and nutritional characteristics of almonds (Prunus dulcis (Mill). DA Webb) as influenced by harvest time and cultivar. Journal of the Science of Food and Agriculture, 98 (15), 5647-5655.

Material and methods

Line 65 - how long were the almonds stored before the analyzes?

Line 68. Analysis of proximate composition - please provide the numbers of standards or other documents, sources of the methodology, according to which these determinations were carried out. As described later in this chapter As described later in this chapter

Results

Line 281 - please replace "Table 1" with "Table 2".

The obtained results should be discussed with new works, with the results presented in recent years (examples are given above).

Author Response

Reviewer 3

Manuscript ID: horticulturae- 1720000

Title: A detailed comparative study on some physicochemical properties, aromatic composition, fatty acid, and mineral profile of different almond (Prunus dulcis L.) varieties

Authors: Okan LEVENT

 Review of the manuscript

Comments

1) The manuscript presented for revision is interesting, but the results and conclusions are not very original. Similar works and studies have been reported in recent years (e.g.  Kodad, O., Socias, and Company, R., & Alonso, J. M. (2018). Genotypic and environmental effects on tocopherol content in almond. Antioxidants, 7 (1), 6;

Özcan, M. M., Matthäus, B., Aljuhaimi, F., Mohamed Ahmed, I. A., Ghafoor, K., Babiker, E. E., ... & Alqah, H. A. (2020). Effect of almond genotypes on fatty acid composition, tocopherols and mineral contents and bioactive properties of sweet almond (Prunus amygdalus Batsch spp. Dulce) kernel and oils. Journal of Food Science and Technology, 57 (11), 4182-4192;

Ibourki, M., Bouzid, H. A., Bijla, L., Aissa, R., Ainane, T., Gharby, S., & El Hammadi, A. (2022). Physical fruit traits, proximate composition, fatty acid and elemental profiling of almond [Prunus dulcis Mill. DA Webb] kernels from ten genotypes grown in southern Morocco. OCL, 29, 9;

Simsek, M., Gulsoy, E., Yavic, A., Arikan, B., Yildirim, Y., Olmez, N., ... & Boguc, F. (2018). Fatty acid, mineral and proximate compositions of various genotypes and commercial cultivars of sweet almond from the same ecological conditions. Appl. Ecol. Environ. Res, 16 (3), 2957-71.

Reply: In the current research, of course some of the results could be reported by different researchers. We compared different origin of almonds in details and compare them our Turkish genotypes and reported significant variations among the samples in terms of studied parameters.

I have a few comments about the manuscript:

 Introduction:

2) Line 28. Please compare the information on the volume of almond production in recent years with the FAOSTAT data.

Reply: The updated FAOSTAT results for the almond production in 2020 was provided and cited as suggested. USA was the leader again.

3) Line 31-41: When discussing the nutritional and health benefits of almonds, please refer to the new articles e.g. Kalita, S., Khandelwal, S., Madan, J., Pandya, H., Sesikeran, B., & Krishnaswamy, K. (2018). Almonds and cardiovascular health: A review. Nutrients, 10 (4), 468.

Mathpal, D., & Rathore, G. (2021). An analysis of health benefits of almonds. ACADEMICIA: An International Multidisciplinary Research Journal, 11 (11), 903-910.

Summo, C., Palasciano, M., De Angelis, D., Paradiso, V. M., Caponio, F., & Pasqualone, A. (2018). Evaluation of the chemical and nutritional characteristics of almonds (Prunus dulcis (Mill). DA Webb) as influenced by harvest time and cultivar. Journal of the Science of Food and Agriculture, 98 (15), 5647-5655.

 Reply: The suggested papers by the reviewer is used in the manuscript.

Material and methods

4) Line 65 - how long were the almonds stored before the analyzes?

Reply: After harvesting, they were kept at room temperature for about 1 month and analysed. It is added to the proper place in the text.

5) Line 68. Analysis of proximate composition - please provide the numbers of standards or other documents, sources of the methodology, according to which these determinations were carried out. As described later in this chapter

Reply: It is added as suggested.

 Results

6) Line 281 - please replace "Table 1" with "Table 2".

Reply: It is revised as suggested.

 The obtained results should be discussed with new works, with the results presented in recent years (examples are given above).

Reply: It is performed as suggested.
